# The Radioactivity of Thorium Incandescent Gas Lantern Mantles

Halmat Jalal Hassan [1,2], Suhairul Hashim [1,3,*], Mohamad Syazwan Mohd Sanusi [1], David Andrew Bradley [4,5], Abdullah Alsubaie [6], Rafael Garcia Tenorio [7,8], Noor Fitriah Bakri [9] and Rozman Mohd Tahar [9]

1   Department of Physics, Faculty of Science, Universiti Teknologi Malaysia, Johor 81310, Malaysia; halmat.hassan@univsul.edu.iq (H.J.H.); mohamadsyazwan@utm.my (M.S.M.S.)
2   Department of Physics, College of Education, University of Sulaimani, Sulaimani 46001, Iraq
3   Ibnu Sina Institute for Scientific and Industrial Research (ISISIR), Universiti Teknologi Malaysia, Johor 81310, Malaysia
4   Centre for Applied Physics and Radiation Technologies, Sunway University, Selangor 47500, Malaysia; d.a.bradley@surrey.ac.uk
5   Department of Physics, University of Surrey, Guilford GU2 7XH, UK
6   Department of Physics, Taif University, P.O. Box 11099, Taif 21944, Saudi Arabia; a.alsubaie@tu.edu.sa
7   Department of Applied Physics II, ETSA, University of Seville, 41003 Seville, Spain; gtenorio@us.es
8   Centro Nacional de Aceleradores, CNA, University of Seville-J. Andalucía-CSIC, 41092 Seville, Spain
9   Atomic Energy Licensing Board, Selangor 43800, Malaysia; fitriah@aelb.gov.my (N.F.B.); rozman@aelb.gov.my (R.M.T.)
*   Correspondence: suhairul@utm.my; Tel.: +60-13-7564706

**Abstract:** The use of thorium in providing the intense white luminescence emitted from gas mantles, has a history of some 130 years, the initial application pre-dating by several decades large-scale urban electric lighting. Accordingly, the thoriated gas mantle has proved itself to be of enormous utility, remaining popular in more rural areas well into the 20th century, continuing to enjoy use in campsites and street night markets lanterns until today. The discovery of thorium in 1828 preceded the discovery of radioactivity, with subsequent little appreciation initially of any potential harm from exposure to radioactivity. Study has been made herein of small quantities of five different types of the thoriated gas mantle, all purchased online devoid of any control measures. Several approaches were used concerning the $^{232}$Th activity and dose consequence. First, the activity of $^{232}$Th was estimated using an HPGe detector, with sample M5 providing the greatest activity at $1.25 \times 10^4$ Bq, exceeding the exemption limit for thorium in a mantle. Compared to sample M5, samples M1 to M4 were low in radioactivity, from $5.1 \pm 1.31$ to $16.33 \pm 1.92$ Bq. Moreover, the thorium content in M5 constituted 50% of the mantle mass, somewhat greater than previous literature values. The dose equivalent rate on the surface of a single M5 mantle was found to be 0.68 μSv/h, while at the surface of a pack of six the level was 1.9 μSv/h. Monte Carlo simulation codes have been used to obtain organ equivalent and effective dose rates, the greatest close contact (10 cm) exposure to an unlit mantle being to the thymus, at 0.68 μSv/h and 0.62 μSv/h for a male and female phantom respectively. Accordingly, with packages of thoriated gas mantles potentially giving rise to non-negligible equivalent doses, greater incorporation of controls on the sale of such items in national radiation protection legislation would seem worthy of consideration.

**Keywords:** gas lantern mantle; radioactive consumer products; Monte Carlo simulation

## 1. Introduction

Since 1885, gas lantern mantles containing the unstable element thorium have been used for indoor and outdoor lighting, thoriated gas mantles producing intense white light luminescence [1,2]. Typically in the manufacture of thoriated gas lantern mantles, rayon fibers are dipped into a nitrate solution formed of an active component containing 99% thorium and 1% cerium [3]. On pre-burning of the gas mantle, the thorium and cerium in the mantle are transferred to thorium oxide and cerium oxide, providing the basis of the brilliant white light emission at burn temperatures [4–7].

Thoriated gas lantern mantles are considered radioactive consumer products, NORM (Naturally Occurring Radioactive Materials) added. NORM are radioactive materials that can be found in nature, primarily the primordial radionuclides $^{238}$U, $^{232}$Th and $^{40}$K [8]. The thorium content in the mantle represents a potential hazard from exposure to ionizing radiation [8]. A number of researchers have studied the radiation risk arising from the thorium contained in lantern mantles [9–12], Furuta, Yoshizawa [7] finding no differences in brightness in the use of thoriated gas mantles and competitor non-radioactive mantles. Even so, the thoriated mantle finds continued use in many countries, typically in the absence of any radioactive information on the mantle packing. Among the purveyors of such items little effort seems to be made in indicating the availability of choice between the use of thoriated and non-thorium based mantles.

From a review of the literature concerning consumer products containing radioactive substances in the European Union, Shaw et al. [13] reported that in much of the developed world the sale of the radioactive gas mantle is either prohibited or subjected to licensing. As instances, European states prohibiting the importation of thoriated mantles include the Netherlands, Italy, Greece, and Switzerland [13]. Conversely, in Sweden, Denmark, Spain, Lithuania, and Norway, mantles are subject to licensing by regulatory bodies, the activity of thorium potentially exceeding the exemption limit 1 kBq [14–16]. In Germany and Finland, radioactive mantles are available to the public [13], a situation also found elsewhere, such as in Malaysia where thoriated gas mantles are freely available without control, including lack of demand for testing, handling, and disposal.

Nowadays, gas lantern mantles are usually used in rural areas and for camping and night markets, as night lights [16,17]. In Malaysia, the country of the present study, many types of the mantle are available for purchase online, also being widely used. In regard to radiation risks to health, as a minimum, it would seem necessary to make an evaluation of the thorium content in these types of mantle, also assessing the risk to users and sellers of these items. The gamma-radiation emitted from the mantles and the possible inhalation or ingestion of the fine thorium oxide powder during replacement operations could represent a health hazard for regular users [6].

The present study seeks to investigate the thorium content of gas lantern mantles currently available in the Malaysian market. We have also performed radiometric characterization and analysis, examining radiological implications arising from contact with packaged thoriated gas mantles. In particular, such risks might be assumed to be greatest for purveyors storing these items in bulk amounts. The research includes estimating organ equivalent and effective dose rates, values obtained through the use of Monte Carlo simulation.

Given the absence of Malaysian regulations on NORM, with gas mantles being sold devoid of details of the level of radionuclide, this study also seeks to influence the existing radiation protection guideline document (LEM/TEK/69) established by the Atomic Energy Licensing Board (AELB) [18], the particular document having the intent to address such issues.

## 2. Materials and Methods

### 2.1. Sampling

Via the online market, five types of the thoriated mantle were purchased, the mantles packaging providing no information concerning the radionuclide content within (see, for instance, Figure 1). In some cases, the mantles are sold in bulk in the absence of packaging. The samples were classified into five batches according to manufacture batch numbers. Two samples from each batch number were subjected to investigation.

The three methods of investigation used were as depicted in (Figure 2): Direct, immersed in nitric acid, and measurement of the residual ash following burning for 1 h.

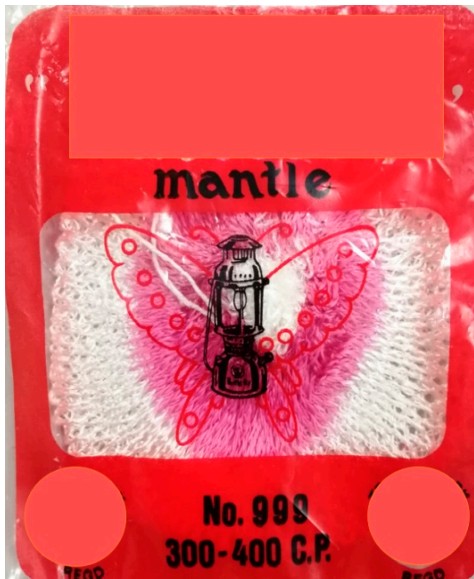

**Figure 1.** Front side packing of the thoriated gas mantle.

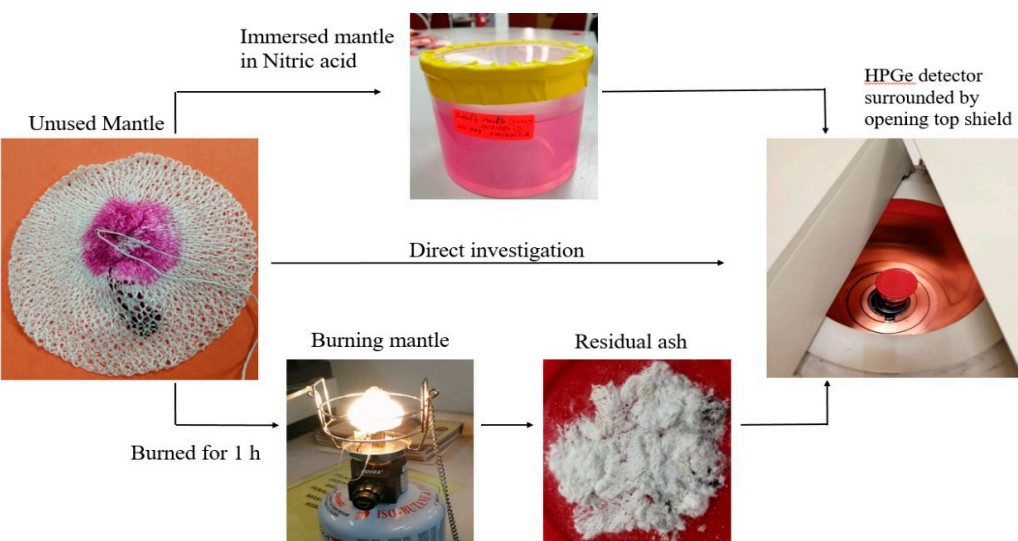

**Figure 2.** Three methods of mantle sample investigation: Direct; immersed in nitric acid; burned for 1 h.

### 2.2. Direct Measurement of Thorium Daughters, Using HPGe γ-Spectrometry

Individual samples of each of the five types of gas lantern mantle were carefully weighed to a sensitivity of 1 mg, sample masses being within the range 1.320 to 3.490 g, obtaining representative mass evaluations. Using a well-shielded high purity germanium (HPGe) spectrometer, each mantle was then counted for a period of 86,400 s (24 h). In detail, each mantle was firmly located within the shielded arrangement, in a holder coaxially aligned with and 3 cm above the top cap of the HPGe detector. This geometrical configuration was maintained for all unburned and burned mantles. The gamma emission from samples was directly measured using an ORTEC GEM Series P-type coaxial HPGe spectrometer (GEM20-76-LB-C-SMPCFG-SV-LB-76; 33% relative efficiency; 1.8 keV FWHM at 1332 keV). This provides for high-performance gamma spectroscopy over the energy range 40 keV up to several MeV. The system, equipped with a Mobius cooling system, also uses high-resolution gamma spectroscopy software (VISION version 8) for spectral analysis of the gamma emission.

The spectra, collected over 16,380 channels, were calibrated with a $^{152}$Eu standard point source, providing a wide range of photon energies (121.78, 244.6, 344.3, 411.1, 778.9, 867, 964, 1112, and 1528 keV). The counting efficiencies of the $^{232}$Th series under these measurement conditions was 1.56% for $^{212}$Pb (238.4 keV), 1.26% for (338.2 keV), 0.44% for $^{228}$Ac (911 keV), 0.61% for $^{212}$Bi (727 keV), 0.75% for (583 keV), 0.17% for $^{208}$Tl (2614 keV), all the energy lines having high emission probabilities of decay gamma rays emitted by the radionuclides [19]. The radioactivity in the mantles were estimated by measuring gamma-ray emissions from the thorium daughters, comprising $^{228}$Ra estimated from the average of the results of both gamma lines of $^{228}$Ac (338 keV and 911 keV), $^{228}$Th estimated from the average of the resulting gamma lines of $^{212}$Pb (238 keV) and $^{212}$Bi (727 keV); also $^{208}$Tl (583 keV and 2614 keV), with $^{232}$Th estimated from the average of $^{228}$Ra and $^{228}$Th.

### 2.3. Mantles Immersed in HNO₃ and Distilled Water

For this, each mantle was immersed in 50 mL of fresh warm 0.5 M HNO$_3$, subsequently stirred for 10 min using a hot plate stirrer operating on the basis of a Teflon coated magnet. Through decantation, the leach solution in each case was transferred to a volumetric flask. The leaching process was repeated using distilled water instead of HNO$_3$. The leach solution was then combined with the previous HNO$_3$ leach solution [17]. To thoroughly leach the radioactivity from the mantle, both the HNO$_3$ and distilled water leaching processes were repeated three times. The mantles were then removed. The combined leaching solution, diluted to 500 mL and transferred into a 500 mL Marinelli beaker, was then measured by an HPGe detector for 86,400 s as previously described. To allow for secular equilibrium, $^{228}$Ra and $^{228}$Th were measured after 30 days of storage.

In regard to a test of manufacturing quality, as required in Indian regulations [20], two unused gas mantles, were immersed in beakers of distilled water maintained at 50 °C through the use of a hot plate, being held at this temperature for 5 h. The mantles were then removed and the radioactivity released into water measured.

### 2.4. Burned Gas Lantern Mantle

For this, the mantle was placed in a lantern and pre-burned for 1 h. The residual ash was then carefully extracted from the lantern and counted for 2 h using the HPGe detector. To estimate the amount of radioactivity in the vapor produced by the mantles during burning, the mantles were measured before burning and 1 h after burning. Further, to observe the build-up of daughter nuclides, the residual ash was investigated 4-, 168-, 240-, and 720 h after burning.

### 2.5. Characterization of Gas Lantern Mantle

For elemental content, the lantern mantles were analysed using an inductively coupled plasma optical emission spectroscopy facility (ICP-OES, Avio 200, PerkinElmer, Waltham, MA, USA), in particular obtaining the amount of thorium in the mantle. For this, 1 g of each mantle was transferred into a microwave digestion vessel to which was then added 10 mL of conc. HNO$_3$ 65% and 5 mL of H$_2$O$_2$ 30%, then heated to 150 °C in a microwave oven to digest the sample. The mixture was evaporated to near dryness, the completely dissolved sample then being transferred quantitatively to a volumetric flask and diluted to 25 mL using ultra-pure water (UPW). Each dissolved sample was then analyzed via use of the ICP-OES facility, the procedure being in accord with manufacturer-defined procedures [7].

### 2.6. Monte Carlo (MC) Simulation and Evaluation of Effective Dose (ED)

Here one seeks organ dose conversion factors (DCFs), estimating organ equivalent doses from gas lantern mantle radiation exposure [21–23]. For this, a series of simulations were undertaken involving the five main series of gamma photons from $^{232}$Th, use being made of the Monte Carlo N-Particle radiation transport code, version MCNP5 (Los Alamos National Laboratory), also involving medical internal radiation dose (MIRD) mathematical phantoms, male and female (Figure 3) (male phantom 178 cm tall, weight 91 kg; female

phantom 168 cm and 72 kg). The mantle was taken to be a sphere of 4 cm diameter. In evaluations the mantle M5 was located at 10, 20, 50, and 100 cm separation from the chest in order to estimate the effective dose. Further to this, organ equivalent doses for 21 organs of the male and female phantom were obtained, with the mantle simulated at 20 cm from the chest.

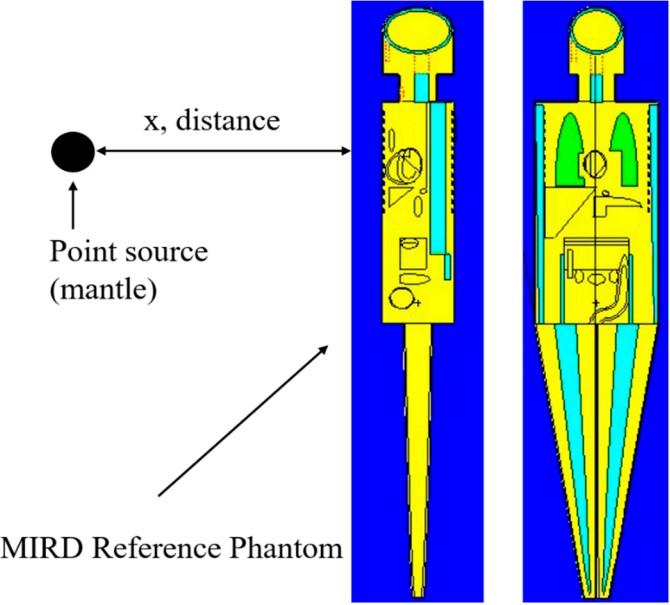

**Figure 3.** Medical internal radiation dose (MIRD) human phantoms and juxtaposition of lantern mantle M5, color represent different composition and density of tissues; green, yellow, and aqua represent lung tissue, soft tissue, and bone, respectively.

## 3. Results and Discussion

### 3.1. Measurements of $^{232}$Th Activity in Gas Mantle

The activity of thorium was investigated for the five different types of mantle (masses 1.320 to 3.490 g), with $^{232}$Th activity ranging from 5.1 ± 1.31 to 16.33 ± 1.92 Bq/mantle. Samples M1 to M4 were low in radioactivity, compared to mantles M5 which recorded a relatively high $^{232}$Th activity, at 12,517 ± 1173 Bq/mantle (Table 1). In addition, the activity of thorium in samples M5 exceeds guidance on an exemption limit of $1 \times 10^3$ Bq for Th-232, as in the international basic safety standards for protection against ionizing radiation IAEA [24], also in update IAEA 2014 GSR Part 3 [15] of $1 \times 10^4$ Bq. Shabana et al. [17] suggest the update links to research showing many mantles exceeding prior guidance, with for instance the authors obtaining $4.5 \times 10^3$ Bq of Th-232 in mantles, also with the U.K. the National Radiological Protection Board (NRPB-R263) proposing a limit of radioactive thorium of not more than 1 kBq per mantle [25].

**Table 1.** Activity of radionuclides of the Th-232 series in different types of the mantle.

| Sample | | Weight (g) | Activity ± 1 Sigma (Bq/Mantle) | | |
|---|---|---|---|---|---|
| | | | **Ra-228** | **Th-228** | **Th-232 *** |
| M1 | Butterfly ART No.4D (500-600 C.P.) | 2.79 ± 0.0027 | 14.11 ± 2.2 | 12.5 ± 1.35 | 13.3 ± 1.77 |
| M2 | Kovea KL 101/102 -TKL929 | 1.42 ± 0.0014 | 6.65 ± 0.6 | 8.6 ± 1.2 | 7.62 ± 0.9 |
| M3 | U-shape ** | 1.32 ± 0.0013 | 4.7 ± 0.73 | 5.5 ± 1.9 | 5.1 ± 1.31 |
| M4 | Kovea TKL-N894 KL103 | 1.75 ± 0.0017 | 14.11 ± 2.24 | 18.56 ± 1.6 | 16.33 ± 1.92 |
| M5 | Butterfly No.999 (300-400 C.P.) | 3.49 ± 0.0034 | 13,393 ± 1398 | 11,642 ± 948 | 12,517 ± 1173 |

* Activity of Th-232, averaged from Ra-228 and Th-228. ** Name of manufacturer and model not available on the mantle package.

Present study results as in (Table 1), are comparable to the literature review data in (Table 2), in the case of M5 recording a value greater than that of the previous data.

**Table 2.** Activity of radionuclide Th-232 in the gas mantle. Literature data and data from this study.

| No. | Th-232 (Bq/Mantle) | Reference |
|---|---|---|
| 1 | 0.56–4.8 | [4]—Iran |
| 2 | 2411 | [5]—US |
| 3 | 247 | [6]—Netherland |
| 4 | 750–1800 | [10]—Italy |
| 5 | 248–893 | [12]—Kingdom of Saudi Arabia |
| 6 | 483–2025 | [11]—Kingdom of Saudi Arabia |
| 7 | 350–4560 | [17]—Kingdom of Saudi Arabia |
| 8 | 1410 | [7]—Japan |
| 9 | 1000 | [13]—Norway |
| 10 | 500–4000 | [13]—Germany |
| 11 | 795–1054 | [14]—Austria |
| 12 | 1386–1963 | [16]—Spain |
| 13 | **5.1–12,517** | **Present study—Malaysia** |

Results from pristine M5 mantles prepared using two different methods (Table 3) showing the activity of Th-232 obtained in direct measurement of the gamma-ray spectrum from 'as is' samples to be comparable to within 11% with that from mantles dissolved in 0.5 M $HNO_3$, indicative of a good degree of efficiency in thorium released into solution.

**Table 3.** Activity of radionuclides from the Th-232 series in mantle M5.

| Sample | Activity $\pm$ 1 Sigma (Bq/Mantle) | | |
|---|---|---|---|
| | Ra-228 | Th-228 | Th-232 * |
| Pristine mantle | | | |
| | 13,252 $\pm$ 1385 | 11,325 $\pm$ 875 | 12,288 $\pm$ 1130 |
| | 13,534 $\pm$ 1411 | 11,959 $\pm$ 1021 | 12,746 $\pm$ 1216 |
| Average | 13,393 $\pm$ 1398 | 11,642 $\pm$ 948 | 12,517 $\pm$ 1173 |
| Mantle dissolved in 0.5 M $HNO_3$ | | | |
| | 11,853 $\pm$ 751 | 10,646 $\pm$ 506 | 11,249 $\pm$ 62 |
| | 11,432 $\pm$ 822 | 10,734 $\pm$ 511 | 11,083 $\pm$ 667 |
| Average | 11,642 $\pm$ 787 | 10,690 $\pm$ 499 | 11,166 $\pm$ 643 |

* Activity of Th-232, averaged from Ra-228 and Th-228; Name of manufacturer and model not available on the mantle package.

### 3.2. Build up Daughter Nuclides

To estimate the amount of radioactivity vaporizing in the burning of the mantles, gamma emissions from the mantles were measured before and after burning, with results recorded in Figure 4; zero-hour represents the unburned mantle while the 1, 4, 168, 240, and 720 h temporal points represent the time delays between initial and post-burn measurements. Subsequent to the initial ignition and termination period, daughter nuclides fractional evaporation is observed, detected at the first-hour temporal point, the initial concentrations of $^{212}$Pb ($t_{1/2}$ 10.6 h), $^{212}$Bi ($t_{1/2}$ 1.01 h), and $^{208}$Tl ($t_{1/2}$ 3.05 m) decreasing by 30%, 37% and 43%, respectively. At 168 h, the build-up of the daughter nuclides restores the initial concentrations, the residual ash retaining the greater thorium fraction. The result accord with previous studies of Luetzelschwab and Googins [5] and Al-Jarallah et al. [11] and Furuta et al. [7], each concluding that thorium is subject to minimal evaporation during burning.

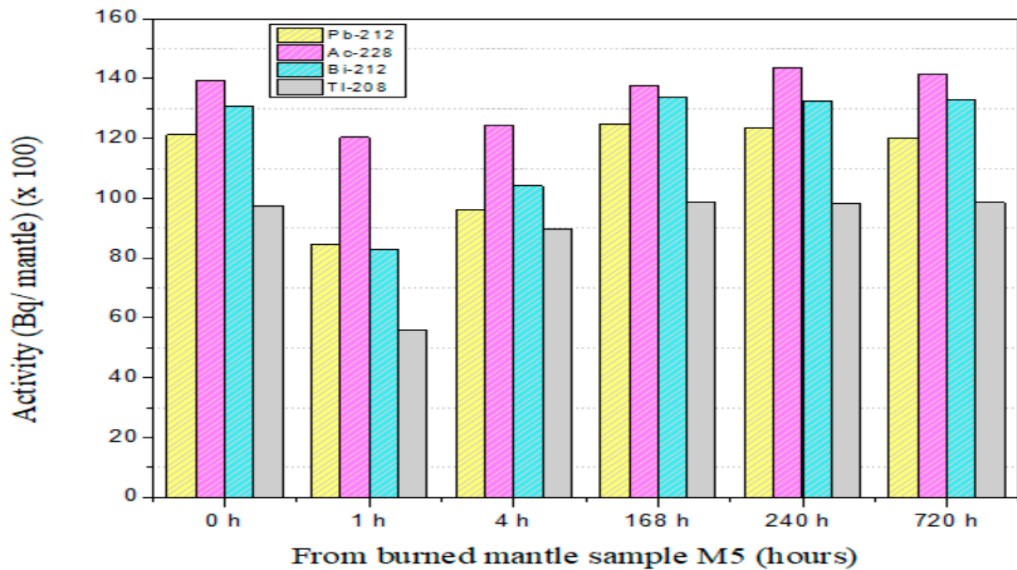

**Figure 4.** Build-up daughter nuclides during and after burning mantle.

### 3.3. Thorium Concentration in Gas Mantle

Thorium concentrations in mantles M1 and M5, analyzed by ICP-OES (Table 4), show values of 1.50 and 507.9 mg/g, respectively. The mantle M5 thorium content of 50% compares with the Furuta et al. [7] review of between 3.1 to 18% and Aksoy et al. [12] of 3.6 to 18.2%. The total absolute thorium content (activity) of the two samples M1 and M5 were 4.05 mg (16.5 Bq) and 1773 mg (7198 Bq), respectively. It is observed that the results for sample M1 and M5 in Tables 1 and 4 are comparable, however the activity for sample M5 (Table 1) recorded values greater than the result in Table 4, due to the high concentration of thorium in mantles M5.

**Table 4.** ICP-OES analysis for thorium content in the mantle.

| Sample | Candle Power (C.P.) | Element | (mg/g) | (mg/Mantle) |
|--------|---------------------|---------|--------|-------------|
| M1 | (500–600 C.P.) | Th | 1.50 | 4.05 [1] |
| M5 | (300–400 C.P.) | | 507.9 | 1773 [2] |

[1] Weight of the mantle M1 (2.7 g), [2] Weight of the mantle M5 (3.49 g).

According to the Indian Atomic Energy Regulatory Board AERB-SS-4 consumer products guidelines, the exemption limit for the content of thorium is 600 mg for candle power up to 400 CP/mantle [20], candle power (CP) representing measurement of luminous intensity. By that measure, the M5 thorium content of the present work is 1773 mg/mantle, a value exceeding the Indian exemption limit. In regard to a test of manufacturing quality, as required in Indian regulations, mantle M1 recorded 8 Bq while mantle M5 recorded 2445 Bq, the latter exceeding the Indian consumer products AERB-SS-4 guidelines, the value for manufacture quality in terms of radioactivity release being limited to no more than 185 Bq. Unlike the AERB-SS-4 provisions in India, in Malaysia no such control is found. More specifically, for present interests in particular, no control measures are found that pertain to the amount of impregnated thorium in a mantle. Moreover, the ability for public purchase conflicts with the ALARA "as low as reasonably achievable" principle, radioactive and non-radioactive mantles being found to be equally bright [7]. The ALARA principle, in the context of the present situation, implies a need for control in the use of the sale and use of such a consumer product.

### 3.4. Organ Equivalent and Effective Doses

Organ equivalent doses were based on MIRD5 mathematical male and female phantoms, with the mantle simulated at 20 cm from the chest. Figure 5 shows the equivalent

doses for the 21 predominant organs, the thymus, heart, and lung being the most greatly exposed organs due to their close proximity to the source.

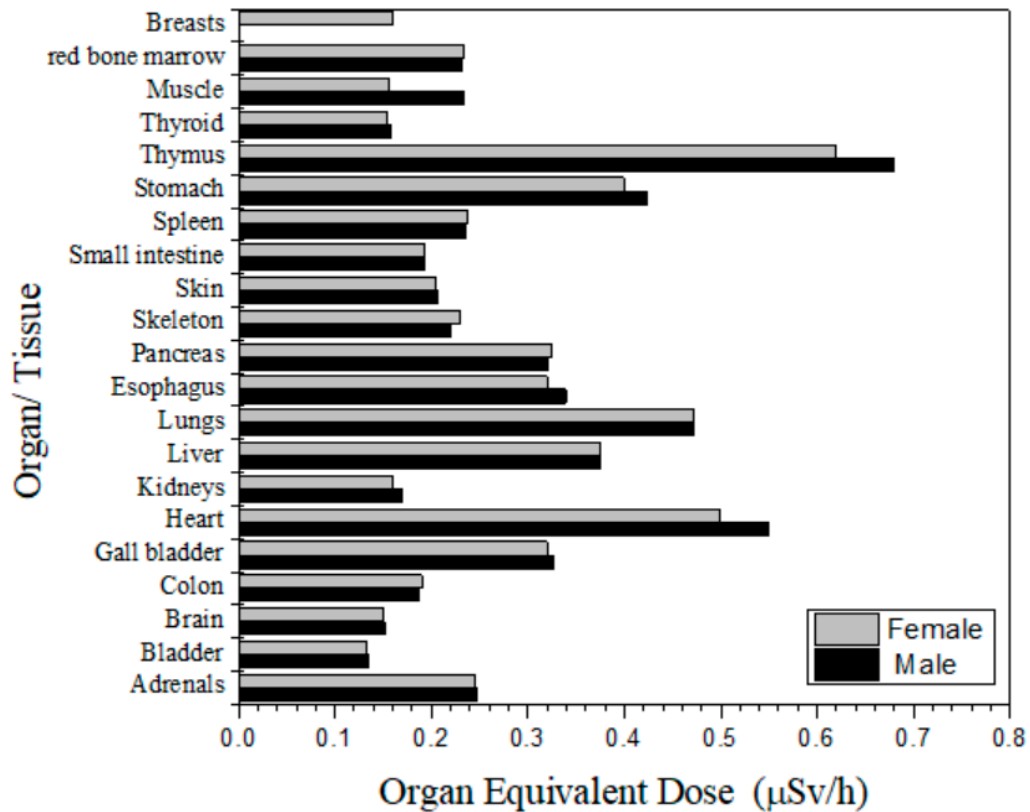

**Figure 5.** Organ equivalent dose (µSv/h) for the 21 organs of the MIRD5 mathematical male and female phantom with mantle M5 simulated 20 cm from the chest.

The greatest equivalent dose resulting from proximity to a single M5 mantle was indicated to be to the thymus, at 0.68 and 0.62 µSv/h for the male and female phantom respectively. Further, the equivalent dose to the stomach is also seen to be greater than that for most other organs due to the larger exposure area. Further investigation has been of the effective dose, as shown in Table 5.

**Table 5.** Effective dose (µSv/h) from mantles at separations from the chest of 10, 20, 50 and 100 cm.

| Sample | 10 cm | 20 cm | 50 cm | 100 cm |
|:---:|:---:|:---:|:---:|:---:|
| M1 | $4.96 \times 10^{-4}$ | $4.67 \times 10^{-4}$ | $1.81 \times 10^{-4}$ | $3.18 \times 10^{-7}$ |
| M2 | $3.87 \times 10^{-4}$ | $3.77 \times 10^{-4}$ | $1.93 \times 10^{-4}$ | $2.27 \times 10^{-7}$ |
| M3 | $2.54 \times 10^{-4}$ | $2.00 \times 10^{-4}$ | $7.60 \times 10^{-5}$ | $1.14 \times 10^{-7}$ |
| M4 | $2.46 \times 10^{-3}$ | $1.87 \times 10^{-3}$ | $8.90 \times 10^{-4}$ | $8.00 \times 10^{-7}$ |
| M5 | $3.41 \times 10^{-1}$ | $2.95 \times 10^{-1}$ | $1.45 \times 10^{-1}$ | $2.94 \times 10^{-4}$ |

With regard to evaluation of the effective dose, the dose conversion factors (DCFs) have been calculated by MC simulation. Table 5 shows the results from a single mantle at separations from the chest of 10, 20, 50, and 100 cm; the effective doses for samples from M1 to M4 are very low to negligible compared to sample M5. It was observed that sample M5 recorded the maximum effective dose, at $3.41 \times 10^{-1}$ at a distance 10 cm, reducing to $2.94 \times 10^{-4}$ at 100 cm.

The dose equivalent was also measured for mantle M5, using a calibrated IdentiFinder 2, FLIR Survey Meter, for exposure periods of 1, 4, 168, and 720 h, the meter being placed at distances of 0, 10, 20, 50, and 100 cm from the mantle. For a single M5 mantle, the contact

dose equivalent was 490 µSv per 720 h (30 days), reducing to 144 µSv at 20 cm (Table 6). For a package of six mantles placed on the surface of the survey meter, a dose equivalent of 1.368 mSv per 720 h was found, reducing to 0.187 mSv at 20 cm. The dose equivalent on the surface of a single mantle M5 was 0.68 µSv/h and on the surface of the package of six mantles, it was 1.9 µSv/h. It was observed that our results in Tables 5 and 6 for single mantle M5 are comparable.

**Table 6.** Dose equivalent (µSv per time exposure) measured with a survey meter (IdentiFinder 2) for mantle M5.

|  | Time Exposure | Source Distance (cm) from Survey Meter | | | | |
|---|---|---|---|---|---|---|
|  |  | Surface | 10 | 20 | 50 | 100 |
| 1 mantle | 1 h | 0.68 | 0.25 | 0.2 | 0.17 | BL * |
|  | 4 h | 2.72 | 1 | 0.8 | 0.68 | BL |
|  | 168 h | 114.2 | 42 | 33.6 | 28.6 | BL |
|  | 720 h | 490 | 180 | 144 | 122.4 | BL |
| 1 pack | 1 h | 1.9 | 0.35 | 0.234 | 0.19 | BL |
|  | 4 h | 7.6 | 1.4 | 0.936 | 0.76 | BL |
|  | 168 h | 319.2 | 58.8 | 39.3 | 31.9 | BL |
|  | 720 h | 1368 | 189 | 187.2 | 136.8 | BL |

BL.*: Background Level (0.1 µSv/h).

In Malaysia, gas lantern mantles are commercially available without restrictions and have been widely used in lighting in rural areas where electricity supply is limited, also for illumination of night food stalls as well as for fishing and camping activities. In the example case of night markets in Malaysia, open from 6:00 p.m. to 10:00 p.m., also with the assumption that a double mantle gas lantern lamp is used for four hours per night, the annual dose can reach 0.6 mSv per year, assuming a dose equivalent at 20 cm of 0.2 µSv/h see (Table 6). This is greatly in excess of the dose constraint of 0.3 mSv/y from any single source [26]. In yet another example, concerning warehouse storage and courier services, workers may receive unnecessary exposures of up to 1.9 µSv/h on the surface of packages of six mantles, referring again to (Table 6). Assuming a worker spends two hours per day handling such packages, the annual dose may exceed the permissible limit of 1 mSv/y for members of the public [27–29]. The standard specifications for consumer products certainly need to be revised for inspection by the regulatory bodies to ensure compliance by lantern mantle manufacturers in order to meet the requirement of keeping the radiation dose to individual members of the public as low as reasonably achievable. It is suggested that such harmonization of practice will provide an important step towards the design and production of safer consumer products for public use.

## 4. Conclusions

Five different types of mantle were investigated in this study. The content of thorium in mantle M5 was found to be 50%, a value exceeding the highest levels reported in the existing literature. Mantle M5 recorded the highest thorium activity at $1.25 \times 10^4$ Bq, exceeding the exemption limit for thorium of $1 \times 10^4$ Bq adopted by the IAEA (2014). Further estimate has shown the dose equivalent on the surface of a single M5 mantle to be 0.68 µSv/h, while on the surface of a package of six mantles it can increase to 1.9 µSv/h. Two scenarios have been described, both showing that close contact with a package of M5 mantles can infer annual doses in excess of the dose constraint of 0.3 mSv/y from a single source, the second example inferring levels exceeding the permissible limit of 1 mSv/y for members of the public. Currently radioactive mantles are available for purchase without information concerning their radioactivity. Additionally, there is no impediment to the purchase of these in Malaysia, the country of present study. There is need to set criteria for the approving of thorium based lantern mantle products before their release for purchase and use by members of the public.

**Author Contributions:** Conceptualization, Methodology, Investigation, Formal analysis, Writing—original draft, H.J.H.; Supervision, Funding, Conceptualization, Investigation, Writing—original draft, Formal analysis, S.H.; Investigation, M.S.M.S.; Writing—review and editing, D.A.B. and R.G.T.; Funding, Conceptualization, A.A., N.F.B., and R.M.T. All authors have read and agreed to the published version of the manuscript.

**Funding:** This work was supported by the Ministry of Science, Technology and Innovation (MOSTI) of Malaysia using Kumpulan Wang Amanah (KWA) Majlis Sains dan Penyelidikan Kebangsaan (MSPK). The authors gratefully acknowledge the Ministry of Higher Education Malaysia and Universiti Teknologi Malaysia through UTM High Impact Research Grant (No. 09G08). This study was also partially funded by contract research grant from Intelligent Science Sdn. Bhd. (R.J130000.7617.4C347). We acknowledge the cooperation with the project partners i.e., International Atomic Energy Agency and Atomic Energy Licensing Board, Malaysia for expert mission and procurement support through IAEA TC programme (MAL9018: Strengthening the Regulatory Infrastructure for Radiation and Nuclear Safety). A. Alsubaie acknowledges Taif University for financial assistance through Project number (TURSP-2020/163).

**Conflicts of Interest:** There is no conflict of interest.

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
