# Peer review of "The Radioactivity of Thorium Incandescent Gas Lantern Mantles"

_applsci, doi:10.3390/app11031311_

Round 1

Reviewer 1 Report

The paper under review is devoted to studies of gas lantern mantles, containing thorium, and the potential radiation damage that the latter may cause, using set of experimental and computational methods. I found this work very interesting, but it should be carefully revised in terms of representation accuracy: figure captions without legends, laboratory devices mentioned without names or models, experimental and discussion chapters are given in a very different styles, etc. I have addressed some questions below, however I think the manuscript should be revised in more detailed level. Thus the major revision is needed.

- line 23: To my knowledge, Th was discovered in 1828.

- I understand that it is not good to advertise products in the scientific paper, but nevertheless, I think that the names of manufacturers and particular models of mantles that were used in the research should be given. It is similar to naming the manufacturer of particular reagent in a “chemical” paper.

- In continuation, please specify the origin of HNO3 acid that was used.

- You have described the gamma-spectrometry, but didn’t say a word about ICP-OES equipment. All devices and methods should be described.

- Chapter 2.6: which mantle was chosen for the experiment – would the results be the same if it would be another mantle from the list of purchased?

- Fig 3 (but also related to other figs): the caption should have a legend for at least the main parts and color scheme.

- Are there any differences in radiation and doses if the burning intensity would change?

Author Response

Subject: Response to reviewers’ comments

Manuscript Title: The radioactivity of thorium incandescent gas lantern mantles

Manuscript ID: applsci-1054185

Thank you very much for your efforts in handling our manuscript. We hereby acknowledge the positive comments and considerations received from reviewers to improve the contents of the manuscript. We gratefully think that the comments have helped us to further enhance and strengthen the paper. The required amendments have been performed, and the revised sections were highlighted in red color throughout the manuscript.

Reviewer 2 Report

SUMMARY: The objective of this paper is to investigate the radiological risk of thoriated gas mantles due to public and occupational exposure to thorium content in such products. Authors suggest that these mantles are still largely used in countries with less developed urban electric lighting, but that also these products are not in some countries under adequate surveillance by regulatory bodies responsible for radiological protection. Therefore, I find this topic not only interesting but also highly relevant.

However, the manuscript itself is not coherently written at places. Also, my impression is that results are highly convincing and interesting, but not elaborated enough. The general impression is the lack of coherence and systematization in Section 2 and Section 3. Therefore, I suggest the authors to revise the following parts of their manuscript and result presentation :

ABSTRACT

  • Authors should provide the whole range of results that they obtained during the study, not only focus on the most radioactive one.

INTRODUCTION

Line 46: phrase “NORM (Naturally Occurring Radioactive Materials) added” is not clear enough. Although I might understand what authors mean by this, someone who is not related to NORM will hardly understand it.

Line 59: Please, specify what these exemption limits. If different countries have different exemption limits, maybe an example for one or two can be given or range of exemption limits. Certainly, specific numbers could make this point more relevant and help to put the problem in the context.

Line 66: Is there any radiological hazard due to inhalation of thoron?

Line 75: Word “radioactivity” might sound too vague for research paper. It is a highly provocative word but usually what we mean by “radioactivity” is radionuclide content and consequent dose rate. At least in radiological protection related to NORM, it is so (e.g. we can talk about NORM waste and residues but hardly never you will hear from radiological protection expert naming such NORM materials as a radioactive). Therefore, I suggest authors be careful about the wording and use more specific terms. This suggestion also goes for line 81.

Also, are we talking about “project” or “study”?

MATERIALS AND METHODS

My biggest objection on this section is lack of coherency and systematics in its presentation (sampling, sample description, sample preparation, method description, data analysis). I suggest the authors to revise this section more systematically.

Line 87: Please provide more info (justification) for the need to measure mantle prepared by three different methods. Direct measurement might be self-explanatory, but readers might be interested in the other two approaches, especially the one that includes nitric acid, the leaching process and how that relates to risk estimation?

Line 95: I suggest adding sample masses that were used for gamma spectrometry.

Line 99: I find this sentence unnecessary in this section (“This varied very slightly for the immersed mantle.”.

Line 108: 911 keV is missing for Ac-228

Line 109: suggestion to add a bit more info bout branching for readers that are not quite familiar with gamma-ray spectrometry

Line 112: Regarding the use of Tl-208, does this means that Tl-208 was estimated from the averages of the results obtained by 583 and 2614 keV lines? Or, that Tl-208 was used to estimate Th-228? If so, how did you account for branching after Bi-212?

Line 115: the subtitle is “2.3 Mantles immersed in HNO3”, but this section describes two types of leaching: distiller water and HNO3!

RESULTS AND DISCUSSION

My general suggestion would be to offer readers activity concentrations (Bq/kg) not only activity/mantle. With activity concentration, it is easier to make comparisons with e.g. usual Th-232 content in the soil or other related NORM products and therefore to put your results in the context.

Yes, activity concentrations can be calculated from the masses you provided, but I do not see why not offering it per se, just for the sake of simplicity.

Also, the low number of samples should be acknowledged by authors (n = 5). Furthermore, the discussion is mostly focused on M-5 samples because of its higher activity. I suggest to authors to reconsider their result assessment and to include other samples too into their consideration.

Line 155: This range includes M1-M4, not M5 sample. Also, “low” in comparison to what?

Line 158-159: Are both of these IAEA publications relevant for the result assessment? If not, use only the newest one as a reference.

Line 167 and Figure 4: I do not think Figure 4 offers anything beneficial to readers. I find it unnecessary. And on the spectrum there is K-40. I do not remember you mentioned this radionuclide earlier in the text. Also, you did not mention Tl-208 at 511 keV or 860 keV, as well as Ac-228 at 968 keV. Does this imply that you have used in your calculations other photopeaks than those reported in Section 2?

Table 2. From this table, it seems that results that you presented the first are for the incandescent mantle, but the caption of Table 1 is not written like that nor are the result at the beginning of this section. Please, explain this.

Table 3. The caption of this table is not informative enough. It seems that there has been only one sample with repeated measurements. But if I understood correctly, it is two samples from the batch M-5.

Line 185 -187: again, why is this important?

Line 191: Comparison of M-1 – M-4 mantles with literature?

Line 209-212: This belongs to the Methods section, not here.

Lines 208 and 217 contain abbreviations C.P and ALARA, respectively, neither explained to readers. It is a general rule that at least abbreviations, upon their first appearance in the text should be explained. But in the context of the study, the importance of the ALARA principle and its implications to this specific problem could be more explained.

Line 220: mantles or residual ash? It is somewhat unclear when intacted mantles were measured when residual ash. Especially since in Table 2 incandescent mantle is mention. Suggestion to authors to consider dividing this section to subsections regarding different types of samples that were measured

Line 234: why 20 cm was chosen as the most appropriate distance? Although this issue should be described in the method section.

Table 5: what does asterisk means at M-1 and M-5?

Line 252: can background level be quantified at least in the footnote of Table 6?

Line 267: what is the reference for this dose constraint?

Line 268 – 269: it is a bit unclear last part of this sentence.

Line 270: again, reference for dose limit.

CONCLUSIONS

Line 274: there were more methods in this research other than gamma-ray spectrometry.

Line 283: Maybe to rephrase this sentence to emphasise that results of this study indicate that there is a need to set criteria…. But generally, in this section more about the study and its result should be said. Too much is said about regulations and consumer product safety. Although, clearly the results of this study imply the need to improve regulation related to thoriated gas mantles in Malaysia.

Author Response

(The authors gave the same response as above.)

Round 2

Reviewer 1 Report

I'm almost completely satisfied with the corrections that have been made to the manuscript. However, I believe one question is still actual. Although figure captions were modfied, the legend is still missing. According Fig 3: what is the color scheme? What does the blue, yellow, green and cyan colors mean? It should be given in the caption's legend.

Author Response

Subject: Response to reviewers’ comments

Manuscript Title: The radioactivity of thorium incandescent gas lantern mantles

Manuscript ID: applsci-1054185

Thank you very much for your efforts in handling our manuscript. We hereby acknowledge the positive comments and considerations received from reviewers to improve the contents of the manuscript. We gratefully think that the comments have helped us to further enhance and strengthen the paper. The required amendments have been performed, and the revised sections were highlighted in red color throughout the manuscript.

Response to Reviewer Comments

Reviewer #1

Comment #1: I'm almost completely satisfied with the corrections that have been made to the manuscript. However, I believe one question is still actual. Although figure captions were modfied, the legend is still missing. According Fig 3: what is the color scheme? What does the blue, yellow, green and cyan colors mean? It should be given in the caption's legend.

Authors reply:

Thanks for your kind observation, critics, and comments, which resulted in the overall improvement of the manuscript.  Please note that Figure 3 provided detailed concern to the color scheme and the manuscript was revised as you suggested for the caption of Figure 3

Line 159-161: Figure 3. MIRD human phantoms and juxtaposition of lantern mantle M5, color represent different composition and density of tissues, green, yellow, and aqua represent lung tissue, soft tissue, and bone, respectively.
